# An Update on Graphene-Based Nanomaterials for Neural Growth and Central Nervous System Regeneration

**DOI:** 10.3390/ijms222313047

**Published:** 2021-12-02

**Authors:** Maria Grazia Tupone, Gloria Panella, Michele d’Angelo, Vanessa Castelli, Giulia Caioni, Mariano Catanesi, Elisabetta Benedetti, Annamaria Cimini

**Affiliations:** 1Department of Life, Health and Environmental Sciences, University of L’Aquila, 67100 L’Aquila, Italy; mariagrazia.tupone@univaq.it (M.G.T.); gloria.panella86@gmail.com (G.P.); michele.dangelo@univaq.it (M.d.); vanessa.castelli@univaq.it (V.C.); giulia.caioni@guest.univaq.it (G.C.); mariano.catanesi@univaq.it (M.C.); annamaria.cimini@univaq.it (A.C.); 2Center for Microscopy, University of L’Aquila, 67100 L’Aquila, Italy; 3Sbarro Institute for Cancer Research and Molecular Medicine, Department of Biology, Temple University, Philadelphia, PA 19122, USA

**Keywords:** nanomaterials, graphene, graphene-oxide, tissue engineering, regenerative medicine

## Abstract

Thanks to their reduced size, great surface area, and capacity to interact with cells and tissues, nanomaterials present some attractive biological and chemical characteristics with potential uses in the field of biomedical applications. In this context, graphene and its chemical derivatives have been extensively used in many biomedical research areas from drug delivery to bioelectronics and tissue engineering. Graphene-based nanomaterials show excellent optical, mechanical, and biological properties. They can be used as a substrate in the field of tissue engineering due to their conductivity, allowing to study, and educate neural connections, and guide neural growth and differentiation; thus, graphene-based nanomaterials represent an emerging aspect in regenerative medicine. Moreover, there is now an urgent need to develop multifunctional and functionalized nanomaterials able to arrive at neuronal cells through the blood-brain barrier, to manage a specific drug delivery system. In this review, we will focus on the recent applications of graphene-based nanomaterials in vitro and in vivo, also combining graphene with other smart materials to achieve the best benefits in the fields of nervous tissue engineering and neural regenerative medicine. We will then highlight the potential use of these graphene-based materials to construct graphene 3D scaffolds able to stimulate neural growth and regeneration in vivo for clinical applications.

## 1. Introduction

Nanotechnology is a multidisciplinary field that involves the manipulation of novel materials at the nanometer scale to obtain specific properties and functions and to design innovative devices suitable for medical science and biology as well as engineering or physics applications.

Those materials that measure between 1 and 100 nm at least for one of their dimensions can be termed “nanomaterials”, and they show remarkable distinction in their physicochemical fundamental properties in comparison with the bulk of the same materials. The shape is crucial to their properties; thus, two different shapes of the same materials can lead to different features. Moreover, the increased surface area per unit mass plays a key role in increasing chemical reactivity by about 1000-fold [1,2].

From this perspective, graphene (G) has been the focus of attention for many years. G is one of the strongest materials [2], a pure crystalline form of carbon with an atom diameter thickness. From the zero-dimensional (0D) fullerene to the one-dimensional (1D) carbon nanotube to two-dimensional (2D) sheets, it is configured in a honeycomb-like structure with sp^2^ hybridized carbon atoms covalently bonded together in the same planar monolayer [3,4]. G has a very large surface area (2630 m^2^ g^−1^ for single-layer G), and every sheet is linked by van der Waals forces with 1TPa of stiffness and 130 GPa of the tensile strength [5].

It shows excellent electrical conductivity, high thermal conductivity, impermeability to liquids and gas, strong mechanical strength, and good biocompatibility [6,7].

High thermal and electrical conductivity results from the atomic structure and electron distribution of graphene, these properties providing extraordinary optical behaviors, excellent mechanical properties, extreme chemical stability, and a large surface area. Due to this evidence, G can perform as nanoscale building blocks to generate innovative structures complexed with biological molecules or other types of nanomaterials with brand new features [8,9].

This versatility to assume different shapes and the possibility to be combined with thousands of molecules makes this material attractive for scientists in the medical field as well as in many other scientific fields. In 2004, for the first time, the G was prepared by mechanical exfoliation of graphite; this approach simplified the obtainment of the material, and its use diffusion has made it very popular in the last decade, although the size and orientation of the obtained sheets are mostly unmanageable with a lateral size up to tens of microns [8,10].

## 2. Graphene and Its Chemical Derivatives

The G derivatives can be classified into two families, namely chemically modified graphene (CMGs) and functionalized graphene (FG). Modification of the fundamental G structure during synthesis or consequent chemical manipulation provides several new chemo-physical features to the material (Figure 1). Leading members of this group are graphene oxide (GO), reduced GO (rGO), and multilayered GO, widely used as the backbone in nanotechnology for countless applications in many fields. CMGs are attractive for their unique properties as well as self-assembly ability, which is a promising approach to obtain advanced materials or more complex systems based on graphene, such as the aforementioned, GO, rGO, and all the other derivatives [11].

CMGs show some advantages compared to pristine G, such as lower cost, easier production at a mass level, better dispersion in solvent (avoiding the stacking between layers), and based on their multiplicity of non-covalent forces, they can easily assembly with themselves or with inorganic and/or organic materials, and working in coherence with them, allow to obtain functionalized graphene and thus multifaceted nanomaterials [12,13].

FG involves covalent or non-covalent modifications, which give additional properties and better performance to these structures. Functionalization can occur with several molecules, such as DNA, RNA, peptide, or much more complex compounds, including anticancer drugs [14]. Other kinds of modification involve the use of nanoscale objects (nanoparticles, nanowires, nanorods, and nanosheets) and polymers [14]. The GO reactive oxygenated groups, such as epoxy, carboxylic acid, and hydroxyl groups can be easily covalent functionalized with small molecules or polymers [15]. Covalent reactions used for GO can be also applied on rGO. Based on GO and G nanosheets, noncovalent interactions can be used to create 3D superstructures, by using aromatic molecules as bridges, linkers, and spacers [16]. Noncovalent functionalization with polymers and molecules requires van der Waals and π–π stacking interactions [17], which do not affect properties such as mechanical strength and electric conductivity, and ensure high flexibility [16].

In this way, it is possible to obtain a variety of composite structures including films, fiber, membranes, or porous scaffolds with exclusive properties, for instance, increased flexibility, surface area, mechanical strength, and electrical conductivity, lighter weight, and gain of functions that ensure CMGs broad application fields [12].

Depending on sizes and dimensions, the fundamental elements for the self-assembled structures can be classified as 0D that include nanoparticles and nanodots, 1D including nanotubes, nanofibers, and nanorods, 2D as nanosheets, and 3D that comprise the complex structure, beads, vesicles, and films [17]. Different dimensionalities are found in different applications in many fields.

Zero-dimensional nanoparticles of magnetic GO coated with poly lactic-co-glycolic acid (PLGA) polymer (NGO/SPIONs) have been used as a dynamic nanocarrier for IUdR (5-iodo-2-deoxyuridine). IUdR induces a radiosensitizing effect but suffers from limited half-life and failure to pass the blood–brain barrier (BBB). G nanoparticles (NPs) improved the penetration of the BBB enhancing the efficacy of glioma therapy [18].

Examples of 1D materials are CMG-based NPs core-shell structures, such as rGO encapsulated SiO_2_, Co_3_O_4_, TiO_2_NPs, Fe_3_O_4_ microspheres. The G cover, wrapped on the NPs (nanosphere, nanowires, and nanotubes), makes these structures suitable for application in electromechanical devices thanks to the conductivity of graphene [17,19,20]. These materials can be used in biosensing; for example, Nayak et al. designed a fiber-optic sensor based on Au-GO to detect the levels of sucrose in biological fluids [21], and it was also shown that Au NPs bound on the graphene nanosheets can be employed in diagnosis to discriminate normal cells from cancer stem cells [22]. Two-dimensional materials can be folded into different shapes and topographies, bending and curving in response to external stimuli. This ability to adapt to large deformations caused by biological forces makes these materials suitable for use in the manufacture of extensible devices or the subject has to change in shape [23]. These 2D materials are also suitable for cell adhesion and growth and can enhance the orientation of fibroblasts on wrinkled substrates. Good viability and a high degree of alignment were shown by cultured mouse and human fibroblasts on wrinkled GO substrates. Moreover, wrinkling or unwrinkling materials could elucidate the cell’s response to topography changes and make it possible to study the timeline of cell differentiation. In addition, wrinkled GO substrates can be used for antimicrobial coating [24,25]. There is recent progress in 2D material fabrication of stretchable nanocoatings that provide skin-mimicking capabilities, such as AgNWs/rGO on polyurethane layer, that exhibited touch sensing capability [26] or an ultra-stretchable GO-latex bilayer, an exceptional barrier that acts as a selective membrane [27]. The intrinsic features of CMGs 2D can act as an optimal template for the synthesis of encouraging new hybrid 2D nanomaterials, loading on the basal plane of graphene and derivatives, organic/inorganic components.

## 3. Biomedical Applications

Beyond the already well-known fields of applications, which include energy technology, nanoelectronics, sensors, and catalysis [28,29], there is an increasing interest in biomedical applications. This is due to the biocompatibility of some G-based materials [30] and their ability to be easily functionalized [31]. Important applications may include smart drug/gene delivery, tissue engineering, optical images, and theranostic. Regarding drug delivery, the reasons behind the use of nanomaterials derive from the necessity to deliver therapeutic compounds into areas of pathosis in a controlled manner without healthy tissues being reached. However, some drugs are characterized by a lower water solubility or an inability to pass barriers, such as the BBB. Many examples from the literature demonstrate the high potential of graphene oxide nanoparticles. Sun and colleagues (2008) developed NGO-PEG (pegylated nano-graphene oxide) sheet-like vehicles, which are able to transport the doxorubicin into cancer cells, in particular pH conditions [32]. GN–CNT–Fe_3_O_4_ (graphene nanosheet–carbon nanotube–iron oxide nanoparticle hybrid) can bind the anticancer drug 5-fluorouracil, and it has been demonstrated that its release is pH-dependent [33]. Some of the advantages reported are related to high specificity and controlled release [34]. The smart gene delivery system enables to transfer DNA to a cell for gene therapy, introducing therapeutic genes for the treatment of monogenic hereditary or acquired disease [35].

However, one of the most promising fields of application is in tissue engineering, which involves the development of biocompatible scaffolds to replace damaged tissue. Bone tissue engineering (BTE) is based on the application of bioactive biomaterials to repair major bone defects, providing support to bone tissue formation; graphene and its derivatives, such as GO, exhibit optimal properties for this application. By designing composite materials, it is possible to produce favorable changes to scaffold morphologies, to mimic the microenvironment topography. For example, it is possible to increase the surface roughness or increase the total porosity and offer better performance in terms of cell adhesion, migration, and differentiation [36,37]. Dinescu et al. realized a bio-construct where human adipose-derived stem cells (hASCs) cells were embedded into the scaffold made of chitosan and different percentages of GO (hASC/CHT/GO). While in the case of control without GO, hASCs maintain round shape and did not achieve a fusiform morphology without long actin filaments, they demonstrate that 0.5% GO adding to the scaffold’s composition, induce the development of actin filaments, 1% GO provides the fusiform phenotype, and a well-developed cytoskeleton was achieved with CHT/GO 3%. The study confirmed that the differentiation of hASCs cells to bone lineage was favored by the presence of GO [37]. Moreover, these considerations were also supported by other evidence that graphene controlled and accelerated the osteogenic differentiation of human mesenchymal stem cells [38]. Dias et al. designed poly(caprolactone) (PCL) composite scaffolds functionalized with few-layer G (FLG) by an extrusion-based 3D printing system. The composite scaffolds provide 40% of porosity that showed good cell viability and long-term proliferation of SaOs-2 cells; thus, it has excellent biological, thermal, morphological, and mechanical properties, and it is an excellent material, ideal for bone tissue regeneration [39]. Furthermore, in this context, in a recent work, poly(propylene fumarate)/polyethylene glycol-modified graphene oxide nanocomposites have been studied. They were successfully produced through sonication and thermal curing. The developed composites showed excellent features in terms of thermal stability, mechanical performance, water absorption, hydrophilicity, hydrolytic degradation, protein absorption capability, cytotoxicity viscoelastic, and antibacterial properties. In addition, cell viability experiments have been performed, demonstrating that PPF/PEG-GO nanocomposites do not lead to toxic effects on normal human dermal fibroblasts (NHDF) and other cell types, suggesting, so far, a considerable potential for medical utilization, principally for bone tissue engineering applications [40,41,42,43].

Great interest in graphene has awakened also concerning the regeneration of skeletal muscle. There are shreds of evidence that natural polymers combined with GO can induce the spontaneous differentiation of myoblasts cells. In particular, some authors used gelatin, collagen, and chitosan, or alginate/GO composite as alginate microbeads, rGO-alginate hydrogels, and bio-printable GO-alginate structures to successfully differentiate C2C12 murine myoblasts cells [44,45]. These studies demonstrated that protein-coated GO particles can increase cell survival, ameliorating cell survival and functionality, inducing the expression of myogenic proteins. GO pro-differentiation effects could be imputable to the ability to adsorb serum proteins and differentiation mediator proteins, especially fibronectin, and to the ability to preserve 3D conformation of cell adherence. A good strategy for scaffold design for this application is graphene foams, an interconnected continuous network of graphene sheets. Some authors show that nickel/graphene foams can induce myotube formation and contraction of C2C12 cells. Foams produced by adding GO to polyurethane (PU) induce spontaneous myogenic differentiation of myoblasts [46].

Ameri et al. used modified graphene with 0.02% gelatin/0.005% fibronectin to analyze HL-1 cells derived from mouse atrial tumors, finding an electrical response in terms of calcium transient measurements that generated the beating characteristics on these cells. This proof of concept makes graphene 3D scaffolds a substrate for culturing cardiac cells [47]. Moreover, Wang and colleagues provided evidence of the role of G sheets in inducing global maturation of cardiomyocytes derived from human-induced pluripotent stem cells [48]. In particular, they observed that a graphene substrate could determine a functional differentiation since cells showed an increase in myofibril ultrastructural organization, electrophysiological properties, elevation in conduction velocity, and Ca^2+^ handling. These changes could be explained by the conductive surface of G, which facilitates electrical propagation, also simulating the suitable microenvironment required [48]. These new bits of knowledge might be useful for applications in cardiac regenerative medicine.

Another field of application of G derivatives is skin regeneration. Electrospun scaffold, made of GO in combination with PLGA and collagen, is a successful treatment for wound healing. Indeed, GO influences the proliferation and migration of dermal fibroblasts, and in addition, is ideal to regenerate different tissue such as blood vessels, connective tissue, and skeletal muscle, providing the appropriate microenvironment [49]. These strategies could provide solutions to diabetic dermopathies. In vivo studies conducted on diabetic and normal rats demonstrated that rGO could accelerate wound contraction, with a significant reduction in wound epithelization time of both normal and diabetic wounds compared to control [50]. Therefore, the 3D scaffold offers a broader strategy for therapeutic research and drug discovery. It is possible to create a biologically active surface to obtain the required conditions where cells can grow in an in vivo-like microenvironment. Indeed, the 3D system offers a tunable substrate that can regulate cellular functions such as gene expression, protein synthesis, growth, differentiation, migration, and apoptosis, via different factors, such as biochemical and electricals but also biomechanical and spatial factors. Factors that trigger the mechanotransduction process could influence the cellular growth in 3D culture, inducing cellular responses that drive changes in cell behavior and morphologies [51,52].

Due to their unique properties, G and its derivatives are considered useful tools in several forms of bioimaging, including optical imaging, magnetic resonance imaging [53], single-photon emission computed tomography, multimodal imaging [54]. Electroencephalography (EEG) and electrocorticography (ECoG) are interesting applications for G that has been applied as a monolayer on the probe used to monitor whole-brain activity or specific brain regions. Due to its extraordinary neural affinity, physical strength, and the ability to interface with living tissue, G can overcome the intense immune response that occurs with other invasive devices, avoiding the activation of reactive glial cells at the implant site, which interrupts the recording channels. These features allow to achieve quantitative and long-lasting recordings of individual neurons for longer periods and recording or provide restoration of functions in the cortex after injuries or neurodegenerative diseases. For this purpose, a G monolayer has been used to wrap 3D intracortical probes, demonstrating its bio-acceptance compared to conventional probes, used in neuroscience, enhancing their performance and reducing their rejection [55].

The 3D structure can be obtained with different strategies, and G can be considered the connective link between nanomaterials and bulk materials. The resulting 3D structures such as foams, sponge aerogel, and hydrogel are very interesting in those fields where surface characteristics are fundamental as in the case of applications in regenerative medicine [56].

## 4. Graphene-Based Nanomaterials for Neural Growth and Central Nervous System Regeneration

The regeneration of injured neural tissues, functional replacement of missing components, and treatment of neurological defects and tumors are just a few examples of the aims. The regeneration of peripheral and central nervous systems is also a big challenge nowadays due to the difficulties of finding safe and effective factors and strategies to stimulate the growth of neurons and repair the damages. The recent progress of studies focused attention on the role of G-based materials in tissue regeneration. The strategies leading to alternative solutions are multiple and include the use of numerous materials as well as different shapes and chemical or biochemical modifications of graphene Figure 2. It is known that, in many cases, it is not enough to use the good shape of scaffold to guide the regeneration, nor is the use of only biochemical stimuli. It is demonstrated that there is a strong connection between the neural circuits and physical extracellular cues. The environment in which the cells grow is fundamental, and topography and nano-topography can influence both the growth and differentiation of neurons as well as mechanical stiffness and electrical signal transmission [57,58]. To this end, new strategies are being developed; one of those is near-field electrostatic printing (NFEP), which merges electrostatic spinning and 3D printing to achieve complex and functional structures. This technique can ensure a well-defined configuration with a preferred topographic pattern, and the fibers are subsequently covered with rGO layers to realize a combined scaffold that provides topography guidance and electrical stimulation. Wang et al. obtained various microfiber patterns from poly(l-lactic acid-co-caprolactone) (PLCL) using NFEP, modulating the fiber diameters, overlay angles, and spatial organization such as web or tubular structure coated with 25–50 layers of rGO. They demonstrated that their scaffold has superior conductivity and inducing an orientated neuronal-like network (PC-12 cells and primary mouse hippocampal neurons) along with the conductive microfibers under electrical stimulation affecting neurite outgrowth [58].

An easy method to obtain an interesting nanotopography, which can be used as a guide for the development of a neuronal network, is an infrared-based photothermal reduction in G with a commercially available laser source under computer software control. This method allows to define, at a submicron-scale, the surface roughness that can promote neuronal adhesion and guides neurite outgrowth and is suitable for different G-based devices [59]. Therefore, nanotopography, as well as the composition of the scaffolds, is fundamental to support and improve the response of neuronal cells in neural tissue engineering applications. We showed that rGO can affect the mechano-transduction in neuronal cells and trigger a switch in signaling pathways [57].

Moreover, another important parameter to take into account is the electroconductivity of the biomaterials in terms of stimulation of regeneration, and rGO plays an important role in this field. The reduction in G can be also made by “green” molecules such as Asian red Ginseng. This molecule can reduce GO (at levels comparable with hydrazine) and the resulting materials can induce differentiation of hNSCs into neurons, more than the GO, due to the higher biocompatibility, higher potentiality in electron transfer, and higher hydrophilicity [60].

The impact of G on the behavior of cells includes many aspects. A study on the axonal transport of the nerve growth factor (NGF), a neurotrophin that is a key player in the axonal elongation, in the presence of G in dorsal root ganglion neurons is reported by Convertino and colleagues. The results showed that G significantly reduces the number of vesicles that retrogradely transported NGF on behalf of a local stall and a raise in axon elongation. The conclusion was that G can have electrophysiological and structural effects that can explain its influence on neuron development and elongation. Neurons can change the charge concentration of G, and this can hyperpolarize the membrane, reducing cell excitability and increasing the axon length by more than 70% compared to cells grown on the control substrate. Moreover, they reported that G induced alteration in the axon morphology such as extended and straighter axonal bundles and nearest and straighter microtubules [61].

Neural stem cells are involved in nervous tissue regeneration, and the main objective is to direct differentiation mainly towards neurons rather than glial cells. With this background, G could act as implantable devices also enabling the process of differentiation. Park and colleagues demonstrated that G substrates could promote human neural stem cells adhesion and differentiation into neurons [58]. In their study, G films were coated with laminins, and cells were seeded in a culture medium containing growth factors, which promote the proliferation phase. Simply by switching to culture media without growth factors, after three weeks, G substrates resulted covered with differentiated cells with visible neurites [62]. Undoubtedly, great interest in the field of regeneration is addressed to the neural stem cells (NSCs), and another successful strategy to enhance NSCs differentiation, particularly towards neurons, is 3D G foam. 3D G foams provide appropriate cell growth microenvironments and guidance cues, in addition to a shift in the regulation of expression of some protein as Ki67, involved in proliferation and the enhancement of differentiation of NSCs, especially towards neurons. Moreover, high interest has been focused on the ability of G foam to induce metabolic reconfiguration of NSCs in relation to a change in the proliferation rate [63] There is evidence that NSC proliferation, induced by G foam, corresponds to a reduction in glycerol that is associated with a change in glucose metabolism, in which high activity can give more energy during proliferation [64]. The reconfiguration of metabolic pathways reflects a different physiological status, and indeed, 263 different metabolites were identified between different growth conditions (2D and 3D G foam), including amino acids, carboxylic acids, organic acids, lipids, cofactors, electron carriers, and others. The 2D and 3D G foam comparison throws a light on the importance of spatial contacts between cell, demonstrating that 3D foam can mimic the in vivo environment very well. In addition, the authors demonstrated that 3D foam increased NSC proliferation compared to 2D G foam and promoted protein digestion and absorption, mineral absorption, biosynthesis of amino acids, and ABC transporter activity. These assets enhanced amino acid incorporation and glucose metabolism, driving the faster NSC proliferation on 3D G foam [60]. Recently, a novel scaffold for culturing NSCs, based on three-dimensional bacterial cellulose-G foam, gave an encouraging three-dimensional substrate for NSC culture [65]. In particular, this kind of substrate is able to increase neurogenesis compared to 3D G foam without cellulose, when NSCs were induced to toward neuron differentiation.

Regarding mesenchymal stem cells (MSCs), recently, by using an electrospun polycaprolactone (PCL) and graphene nanocomposite, a one-step approach able to differentiate this kind of cell into functional dopaminergic neurons has been proposed. The combination of PCL and graphene showed an optimal nanotopography, combining guidance stimuli and substrate cues, and it significantly enhanced the differentiation of mesenchymal stem cells into dopaminergic neurons. These cells exhibited a unique neuronal arborization and enhanced intracellular Ca^2+^ influx and dopamine secretion. Thanks to their characteristics, CL-G nanocomposites can represent a promising model for applications in neural regenerative medicine [66].

Recently, it has been demonstrated that collagen-coated 3D G foams allow the differentiation of mouse MSCs into DA neurons using brain-derived neurotrophic factors. The differentiated cells show positivity to NeuN, β-III tubulin, and TH immunostaining, an increase in neurite extension length compared to cells cultured on other materials, and no cells toxicity have been observed in vitro [67]. These results suggest that these kinds of scaffolds could be a candidate for increasing dopaminergic neuron differentiation in vitro; opening a new window on Parkinson’s disease therapy, although a better characterization of dopaminergic neurons is strongly required to validate the actual differentiation towards midbrain DA neurons and specifically versus A9 phenotype.

Magaz et al. applied GO/silk-based micro/nanofibrous scaffolds in nerve tissue regeneration. The reduction in GO into SF/rGO (silk fibroin/reduced GO) scaffold greatly increased its electrical conductivity and NG108-15 neuronal cells can adhere and survive on the composite. Furthermore, GO can enhance metabolic activity and proliferation as well as sustain neurite outgrowth. The authors hypothesized that the benefits to cells can come from the reduction in GO and the different roughness given by the reduced GO [68]. Thus, it becomes necessary to examine the different cell’s responses to the different states of GO reduction. Investigation on SN4741, a mouse embryonic substantial nigra-derived neural cell line, seeded on fully reduced GO (FRGO), and partially reduced (PRGO) in powder and film revealed that films are more biocompatible concerning powder and, in general, there is a trend towards the proliferation for each substrate analyzed. Taking into account the survival and proliferation of this cell line, as well as differentiation, maturation, and bioenergetic, PRGO displays the best performance. It has been demonstrated that PRGO-films are the most appropriate candidates for the cultivation of DA cells, thanks to their ability to encourage the acquisition of a midbrain DA phenotype in SN4741 cells, without generating unfavorable effects on cell metabolism or mitochondrial function. In particular, impressive modifications in cell morphology were revealed at day 7, observing the acquisition of neuronal morphology as demonstrated by the presence of processes, branches, connections and the expression of neuronal markers, such as Tuj1, TH, and an increase in the level of mature DA neuron markers, such as DAT, GIRK2, synaptophysin and synaptobrevin [69]. Moreover, interestingly, GO treatment was able to induce the expression of specific midbrain transcription factor, such as Lmx1a, Lmx1b, Nurr1 and Pitx3, confirming the specific SN4741 maturation versus neurons characterized by a midbrain DA phenotype [69,70,71,72]. In addition, the maintenance mitochondrial function was supported by PRGO-film and this form also prevents a decrease in the a-synuclein (a-syn) in a Parkinson’s disease model. These findings suggest that PRGO is the most successful state of graphene in promoting DA differentiation and to become a useful scaffold for the study of cell replacement therapy in Parkinson’s disease application [69].

It is worth noting that it is important to consider the glia in tissue regeneration effort because astrocytes play an essential physiological role in the human brain. There is a new strategy to improve the interaction of GO with astroglial cells in the brain, through its chemical modification with a synthetic phospholipid (PL). First of all, the cell adhesion, for example, of primary rat cortical astrocytes, is increased by about three times on GO–PL substrates, necessary in devices used to modulate physiological activity and in addition, GO–PL have been shown to be a good biomimetic approach without significant inflammatory reaction, avoiding glial scar formation after scaffold implants in the brain [73]. Pradhan et al. devised an injectable hydrogel based on graphene oxide and polyacrylic acid functionalized with acetylcholine to repair focal brain injury. The release of acetylcholine is crucial for brain repair of damage that it is characterized by disequilibrium in the cholinergic system. Then, the possibility to release this substance at the site of injury is a great advantage to compensate for the imbalance and, indeed, the study shows the promotion of neurite outgrowth, microtubule networks stability, the expression of neural markers, and an important enhancement of reactive astrocytes in the region of the injury that leads to fast recovery of the damaged brain [74].

Another major challenge is the repair of the central nervous system after spinal cord injury (SCI), which is a debilitating condition for people of all ages [75]. SCI involves the disruption of neuronal axons and the establishment of inhibitory conditions for spinal tissue regeneration. Among the primary strategies, the transplantation of neural stem cells and application of precursors could help to deal with the lack of regenerative capability of the central nervous system. The GO foams (GOFs) also represent an optimal electrically conductive 3D scaffolds and they are suitable to differentiate hNSCs into neurons. It is possible to scale it, for example, rolling GOFs, and use it as a guide to directionally grow neural fibers. The rolled GOFs acquire super-hydrophilic characteristics, and these interfaces together with the porosity of the structure induce proliferation and differentiation of the hNSCs, more evident and accelerated toward neurons instead of glia, under electrical stimulation [76]. Moreover, the in vivo implantation of G foams for spinal cord injury reveals their potential as effective tools for neural repair. It occurs massive cell/protein infiltration after implantation, which provides mechanical stability, extensive vascularization, and colonization by neurites and myelinated excitatory axons of the scaffold. The rGO implants do not show organ toxicity or alter the normal rat behavior but provide positive biological effects when implanted in the injured spinal cord [77]. To repair an SCI, it will be also a good strategy to combine the use of growth factors with tissue engineering techniques. In this regard, insulin-like growth factor 1 (IGF-1) and brain-derived neurotrophic factor (BDNF) was successfully immobilized on GO-PLGA electrospun nanofibres. This approach demonstrates to protect neural stem cells (NSCs) from oxidative stress and to enhance neuronal proliferation and differentiation in vitro. In an animal model of spinal cord injury, the scaffold demonstrates to be able to locally deliver IGF-1 and BDNF that improved functional locomotor restoration and increased the number of neurons at the lesion site [78]. Qian et al. obtained a functional and morphological recovery in a long nerve defect model using GO/polycaprolactone (PCL) nanoscaffolds. In vitro studies confirmed Schwann cell attachment, viability, and proliferation, and even the maintenance of neural characteristics. In vivo performance of GO/PCL scaffolds provided nerve guidance with the repair of a 15 mm sciatic nerve defect. This scaffold exhibits pro-angiogenic features due to the presence of GO [79]. In addition to GO/PCL, Fang et al. used GelMa to obtain nanofiber for guided nerve regeneration to increase the scaffold electroconductivity and biocompatibility. In vitro applications showed the significant enhancement of RSC96 cells (Schwann cell) proliferation and, additionally, this hybrid material can activate the expression of an epithelial-mesenchymal transition (EMT)-related gene. In vivo results demonstrated sensory and motor nerve regeneration as well as functional recovery of treated animals [80]. Moreover, Wang et al. used graphene-based scaffolds under electrical stimulation to investigate how graphene-based nanofibrous scaffolds regulate Schwann cell behavior, as well as on PC12 cell differentiation. Particularly, ApF/PLCL (Antheraea pernyi silk fibroin/(poly(L-lactic acid-co-caprolactone) nanofibrous scaffolds were coated with rGO, which increased the mechanical properties, electroactivity, and biocompatibility of the scaffold. The obtained composite scaffold provided a significantly enhanced proliferation, migration, and myelination of Schwann cells and enhanced myelin-specific gene expression and neurotrophic factor secretion as well as in increasing of PC12 differentiation rate in vitro. Moreover, implantation into rat sciatic nerve defects of the conductive AP/RGO exhibited a healing capacity and promote nerve regeneration in vivo [81]. Therefore, since peripheral nerves show a self-healing potential, successful implantation of the graft to bridge the gap is the best approach. Bioactive materials are fundamental to avoid secondary damage made by the graft, such as the necessity of its replacement after a few times, due to the inflammatory response. Moreover, the intrinsic properties of bioactive materials and their functionalization can provide nerve guidance to repair the gap at the injury site and restoration of function [79,82].

Indeed, one of the major issues in treating neurological diseases is represented by the strategies for brain drug delivery through the BBB. Nanomaterials were demonstrated to pass this physical barrier, acting as a platform through which therapeutic agents could be transported to the brain. In in vivo studies conducted on rats, Mendonça and colleagues demonstrated that the intravenous injection of rGO could down-regulate tight and adhesion junctions and basal lamina proteins, facilitating the passage of these materials. As a consequence, rGO was mainly found in the hippocampus and thalamus of rats [83]. Functionalized graphene can be useful to allow the transport of several biomolecules through the BBB. GO with transferrin [84], graphene quantum dots attached with α-synuclein fibrils and biotin [85], or PEGylated rGO are only a few examples [86] of nanostructures that resulted in successful crossing of the BBB and each of which refers to a particular uptake mechanism and penetration strategy. The possibility to pass through the BBB has important implications in treating neurological disorders or conditions, for which the site-specific administration of drugs is required. In addition, the large surface available on the G platform allows the conjugation of different molecules, such as genes, proteins, antibodies [87]. In vivo studies on glioma investigate the possibility to use G for the nano-delivery of drugs for glioma therapy, demonstrating a major targeting efficiency of the chemotherapic drug such as doxorubicin and 5-iodo-2-deoxyuridine, which are mainly distributed in glioma tissues [18,84].

In the last decade, innovative graphene-based materials have attracted great attention, thanks to their promising physico-chemical properties and their attractive design, characterized by minimal modifications without fully coating/functionalization. A poly(3-hydroxybutyrate) [P(3HB)] matrix with graphene nanoplatelets composites has been developed that is able to enhance the matrix electrical conductivity. In fact, primary cortical neurons plated on these innovative scaffolds showed a significant increase in the neuronal network interplay and activity, suggesting considerable potential for targeting therapies for neural disorders [88]. In an ideal interface to neurons, both high wettability and electrical conductivity are required. In this regard, the same authors showed that hydrogenated graphene (HGr) fosters, characterized by higher wettability respect to graphene, are able to promote neuronal adhesion and network maturation and increase the modulation of neuronal activity. This exciting result suggests the crucial role of wettability in the enhancement of neuronal excitability, more than electrical conductivity [89]. In order to mimic the electrical and physiological environment for nerve regeneration, the use of reduced graphene oxide-coated polycaprolactone fibrous scaffolds, fabricated by electrospinning, revealed a higher level of proliferation and nerve growth factor (NGF) expression of Schwann cells. The use of electrical stimulation mediated by templated scaffolds with aligned topography and electro-activity with rGO loaded can be considered as a proper candidate for a multifunctional scaffold for peripheral nerve injury repair [90]. Moreover, by performing in vivo experiments, graphene oxide-composited chitosan scaffolds were evaluated by neural cell tests for the treatment of traumatic spinal cord injury (SCI). Analyzing the behavioral and electrophysiological results, we can state that the chitosan-graphene oxide scaffold could significantly improve the neurological activity of rats [91]. In order to ameliorate the biocompatibility of graphene, it has been associated with waterborne biodegradable polyurethane (PU) to develop a polyurethane-graphene nanocomposite. Endothelial cells (ECs) and NSCs cocultured on this nanocomposite became more vascular-like and glial-like without the addition of induction culture medium, as demonstrated by an increase in the expression of specific vascular-related and neural-related gene markers, KDR, VE-Cadherin, and GFAP. Interestingly, PU-graphene nanocomposites were also able to stimulate the regeneration of the peripheral nerve in a rat sciatic nerve model [92]. Furthermore, in a recent study, a collagen graphene cryogel scaffold has been proposed. A collagen-based nerve conduit was fabricated using amino functionalized graphene as a crosslinker. Electrical stimulation on collagen graphene cryogels have been shown to affect the BM-MSCs stemness and neuronal differentiation. BM-MSCs grown on collagen graphene cryogels have shown immune-modulatory secretion under an inflamed microenvironment. Interestingly, studies on organotypic culture of the spinal cord demonstrated cellular growth and migration within the collagen graphene cryogel, suggesting a strategy to overcome the major barriers during the spinal cord regeneration [93]. In Table 1, we provide the current and most promising graphene-based nanomaterial applications for neural growth and central nervous system regeneration.

## 5. Crucial Aspects of Biocompatibility and Toxicity Evaluation

Due to the increase in the utilization of graphene-based materials, the assessment of biocompatibility, stability, and toxicity evaluation is a fundamental aspect. In Vivo testing of biocompatibility is a critical step in regeneration approaches. Graphene nanomaterials can be biocompatible or have toxic effects on cells, depending on their chemistry (synthesis, functionalization, surface charges), doses, lateral size, purity, and hydrophilicity. In Vitro evaluation of toxicity required the involvement of macrophages, epithelial or endothelial cells, blood cells, or tumor cells [1]. GO, for example, has been reported to exert toxicity effects towards different human cell lines, including fibroblasts, hepatocarcinoma, skin keratinocytes [104,105,106,107]. However, these effects are strictly related to doses. Studies on human fibroblast cells showed that cytotoxicity and apoptosis induction has been reported at doses >50 μg/mL, while no toxic effects were observed at concentrations <20 μg/mL [104]. The effects of GO on mice have been also evaluated, and the administration of 0.4 mg GO suspension via injection determined chronic toxicity and even mouse death [104].

The damage on humans and tissue can be also caused by the generation of reactive species, such as oxygen-free radicals. This kind of nanotoxicity has also been investigated in the zebrafish model, wherein the administration of GO induced adverse effects on development with an increase in oxidative stress [105]. For G toxicity, it is important to take into consideration not only its dose but also other factors, such as physicochemical properties and the method of administration. For example, among the configuration of G-based materials in the shape of microfibers, recently, in vivo studies have demonstrated that compact, bendable, and conductive rGO microfiber [61] implantation in the injured rat spinal cord was colonized by cells without evidencing signs of acute and chronic local toxicity.

The levels of toxicity of rGO/GO are affected by chemical functionalization, as shown by PEGylated GO, in which the cytotoxicity decreased, as well as in two-dimensional material coated with bovine serum albumin [63]. Another concern is represented by the biodegradability of nanomaterials, which is related to the safety profiles. Biodegradable materials can be degraded in the body in physiological conditions, and they guarantee an absence of long-term toxicity and immunogenic responses. In vitro studies are crucial for understanding the degradation of the scaffolds during long periods. For example, PCL (poly-caprolactone) membranes functionalized with reduced graphene oxide reveal that after one year of hydrolytic degradation, the pH was poorly affected, and the rGO is not mainly released by membranes with a low cytotoxic effect. The mechanical stability of the membranes can be affected by the presence of rGO, but the increase in degradation facilitates cells infiltration and consequently tissue formation. Kurapati and colleagues demonstrated the ability of human eosinophil peroxidase to catalyze the degradation of graphene materials under low concentrations of hydrogen peroxide and NaBr [108]. An approach to overcome the problem of degradation may involve the use of TiO_2_ after photocatalytic reduction or near-infrared light [109,110]. In Table 2, the main advantages and limitations for the utilization of graphene-based materials in regeneration approaches are reported (Table 2). Still, substantial effort is required from researchers and clinicians to develop biocompatible implants that could guide neural tissue repair in the lesion site, and in vivo effects on the nervous system remain to be explored, encouraging findings that indicate that graphene-based nanomaterials have significant potential as novel therapies for neurodegenerative disease.

## 6. Conclusions

G has been employed in a huge field of applications, and the regeneration of nervous system tissue is one of the most complex and attractive, providing the possibility to treat neurological disorders and understand their underlying mechanisms. Repair of central nervous system injury is a big challenge today, and many aspects must be taken into account to design a scaffold based on a biomaterial that can provide support to cell growth and induce tissue regeneration without inflammatory reactions.

Several nanomaterials are suitable for neural growth but do not match all the characteristics required for an ideal scaffold. This limitation can be overcome by functionalizing the biomaterials and creating a composite with a gain of advantageous features.

Graphene and its derivatives are still today the most promising nanomaterials for the regeneration of the nervous system. Indeed, graphene is a novel carbon-based material with extraordinary and unique nanostructure and characteristics. In fact, it is characterized by excellent mechanical, thermal, and electrical properties. Interestingly, it is also able to provide a better replacement for current nanofillers in polymer matrices. Thanks to their properties, graphene-based nanomaterials have shown high versatility, and they have been considered some of the most promising tools in the promotion of tissue regeneration, both for in vitro and in vivo applications. However, due to its limitations already discussed in the previous section, there is an urgent need to examine its mechanism of action in more depth. It is well documented that different formulations of nanomaterials are able to provide very similar results in terms of proliferation and differentiation. At the same time, small changes in scaffold formulation may lead to completely different developments. Accordingly, it would be useful to carry out a systematic study to analyze the repercussion of different nanomaterial compositions on proliferation and differentiation. Scientists researching the roles of each nanomaterial in depth will be able to perform an accurate comparison; as a result, on the basis of the specific characteristics of each scaffold, they will develop a finer rational design.

The biological platforms that can be created with graphene and derivatives, due to their versatility in all its meanings, exhibit their therapeutic potential by exploiting the synergistic effect of different signals, such as topographical and chemical/biochemical cues. The shape, surface area, roughness, and stiffness are crucial to provide not only support to cell growth but to guide differentiation towards a specific lineage, and it is possible to increase these effects using biomolecules to functionalize the G surface to obtain an optimized and ideal device for complete and accelerated tissue regeneration. However, the path leading to the production of commercialized nanomaterials is long and involves exceeding limits presented by the current prototypes.

## Figures and Tables

**Figure 1 ijms-22-13047-f001:**
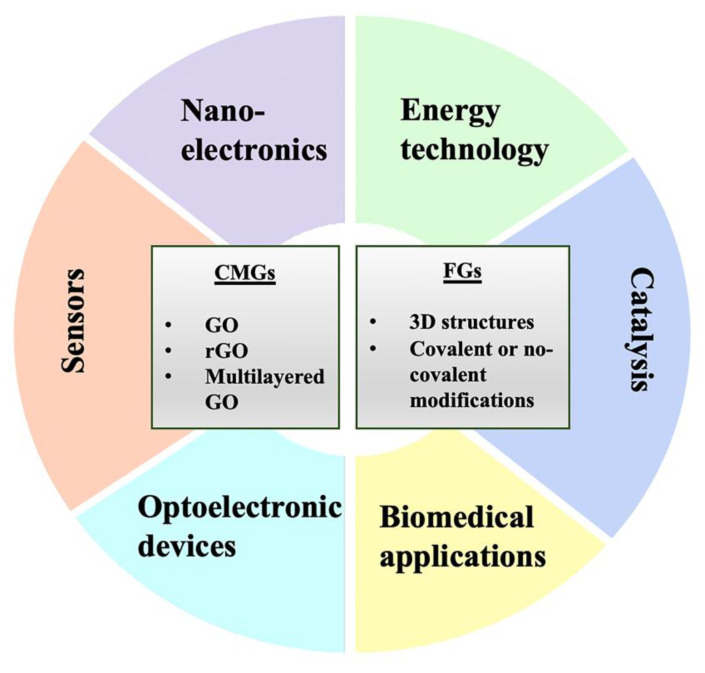
Overview of principal applications of graphene and its derivatives. Chemically modified graphene (CMGs), functionalized graphene (FG), graphene oxide (GO), reduced GO (rGO).

**Figure 2 ijms-22-13047-f002:**
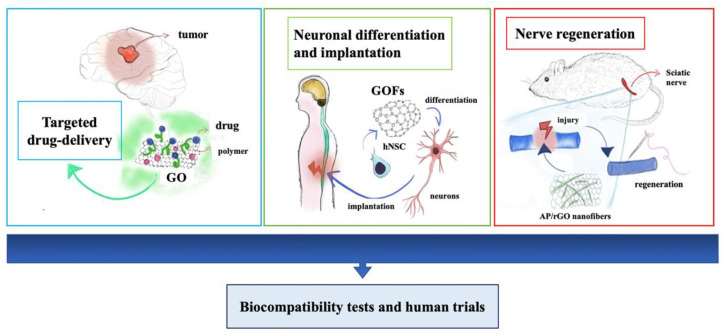
Biomedical applications of graphene-based nanomaterials for neural growth and central nervous system regeneration.

**Table 1 ijms-22-13047-t001:** Current and most promising graphene-based nanomaterial applications for neural growth and central nervous system regeneration.

Graphene-Based Scaffold	Biomedical Applications	Main Results	References
rGO encapsulated on poly (l-lactic acid-co-caprolactone) microfibers (PLCL)	Development of 3D neural networks	Neurite outgrowth and formation of orientated neuronal-like networks	[58]
Laser-Scribed rGO	Generation of micropatterned in vitro neuronal networks	Adhesion and survival of rat primary neurons and, at the same time, guide the subsequent elongation of neurites	[59]
Ginseng-rGO sheets	Neural stem cell (NSC) differentiation	Accelerated differentiation of neural stem cells into neurons	[60]
3D-graphene foam	NSC proliferation and cell fate decision	Enhancement of neural stem cell proliferation through metabolic regulation	[64]
electrospun polycaprolactone (PCL) and graphene (G) nanocomposite	MSCs differentiation	Enhancement of differentiation of MSCs into dopaminergic neurons	[66]
Silk/GO micro/nano-fibrous scaffold	Nerve regeneration	Enhancement of metabolic activity, Neuronoma NG108-15 cells proliferation and neurite outgrowth	[68]
GO, full reduced (FRGO), and partially reduced (PRGO) powder and film scaffold	Neuron differentiation and survival	Promotion of DA differentiation and prevention of DA cell loss	[69]
Choline-Functionalized Injectable GO Hydrogel	Neural regeneration and brain injury repair	Promotion of neurite outgrowth, stabilization of microtubule networks, and enhancement of neural markers expression	[74]
3D porous rGO foams scaffold	Neural repair	Ingrowth of myelinated vGlut2^+^ axons within rGO scaffolds	[77]
GO-PLGA hybrid nanofibres	Spinal cord repair	Enhancement of neuronal proliferation and differentiation in vitro, and NSCs protection from oxidative stress	[78]
GO/Polycaprolactone nanoscaffold	Neurite regeneration	Promotion of functional and morphological recovery in peripheral nerve regeneration	[79]
rGO-GelMA-PCL hybrid nanofibers	Peripheral nerve regeneration	Promotion of both sensory/motor nerve regeneration and functional recovery in rats	[80]
rGO-coated ApF/PLCL (AP/RGO) scaffold	Peripheral nerve regeneration	Enhancement of SC migration, proliferation, and myelination in vitro and promotion of nerve regeneration in vivo	[81]
poly(3-hydroxybutyrate) [P(3HB)]/graphene nanoplateletes composite	Neuronal network development	Promotion of neuronal growth and maturation	[88]
Hydrogenated Graphene	Neuronal regeneration and electrical sensing/recording.	Promotion of neuronal adhesion and network maturation and modulation of neuronal activity	[89]
rGO-coated polycaprolactone fibrous scaffold	Nerve regeneration	Higher level of proliferation and nerve growth factor (NGF) expression of Schwann cells	[90]
Chitosan-graphene oxide scaffold	Nerve regeneration	Recovery of neurological function after spinal cord injury	[91]
Silk/Gelatin scaffold	Nerve regeneration	Increase in neuronal adhesion, differentiation, and neurite elongation	[94]
Polyurethane-Graphene Nanocomposite	Neural tissue engineering	Increase in neurovascular regeneration and peripheral nerve regeneration	[92]
Graphene collagen cryogel scaffold	Neural tissue regeneration	Neuronal differentiation; immune-modulatory secretion; cellular growth and migration on organotypic culture of spinal cord	[93]
Graphene/silk fibroin scaffold	Neural tissue engineering	Neurite outgrowth	[95]
Aminated graphene oxide (NH_2_-GO) scaffold	Nervous tissue regeneration	Induction of neurite elongation and increase in branches in cortical neurons	[96]
Electrospun PCL/gelatin/graphene nanofibrous mats	Nerve tissue engineering	Increase in PC12 cells attachment and proliferation	[97]
N-cadherin-graphene oxide-based scaffold	Neuron development and regeneration	Stimulation of neuronal growth and intracellular transport	[98]
Graphene nanoplatelets (GNPs) and multiwalled carbon nanotubes (MWCNTs) and chitosan scaffold	Neural cell regeneration	Differential neural cell adhesion and neurite outgrowth	[99]
3D-Printed PCL/rGO Conductive Scaffold	Neural tissue engineering	Neural differentiation	[39]
Collagen-coated 3D graphene foam (GF)	Neural tissue engineering	Differentiation into dopaminergic neurons from MSC	[67]
rGOaCNTpega-OPF-MTAC composite hydrogel	Nerve regeneration	Enhancement proliferation and spreading of PC12 cells; stimulation of neurite development	[100]
GOa-CNTpega-oligo(polyethylene glycol fumarate) (OPF) hydrogel	Neural tissue engineering	Increase in electrical conductivity; stimulation of neurite development	[101]
GO and rGO mat	Neural tissue engineering	Neurogenic differentiation	[102]
Graphene-Polyacrylamide Hydrogel	Tissue engineering	Development of synaptic activity	[103]

**Table 2 ijms-22-13047-t002:** Main advantages and limitations for the utilization of graphene-based materials in regeneration approaches.

	Notes	References
**Advantages**		
Biocompatibility of some GBMs	Since they interact with cells, tissue and organs, harmful effects should be avoided.	[1,30,63,82]
Easy functionalization	GBMs can be adapted using covalent or no-covalent modifications and assembled with organic or inorganic molecules	[14,31]
Ability to pass barriers	Graphene nanoparticles can improve the penetration of drugs through BBB	[18,32]
Malleability	Materials can fold in different kinds of shapes and topography	[111]
Application in tissue regeneration	Two-dimensional and three-dimensional structures are suitable for cells adhesion, growth and differentiation, supporting tissue repair	[36,48,49,62]
**Limitations**		
Toxicity of some nanomaterials	Chemical features, functionalization and doses could influence the safety of these compounds.	[104,112]
Biodegradation	The clearance and elimination from the body represent another concern related to biocompatibility and safety, especially for long-term exposure.	[108,109,110]
Route of administration	These compounds exert different degrees of toxicological effects depending on the routes of administration	[113]

## Data Availability

No new data were created or analyzed in this study. Data sharing is not applicable to this article.

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
