# Peer review of "An Update on Graphene-Based Nanomaterials for Neural Growth and Central Nervous System Regeneration"

_ijms, 2021, doi:10.3390/ijms222313047_

Round 1
Reviewer 1 Report
The topic of the article is highly suitable for the journal and it's quite interesting for the scientific comunnity working on carbon-based nanomaterials for biomedical applications. The authors have provided a very brief overview of the current state of art in the field. There are many many articles on that topic so the authors need to carry out a more extensive survey including most works from the last five years and compile them and include them in a table.
Another point is that the authors should critically discuss advantages and disadvantage of these materials for biomedical applications.
Further, articles dealing with polypropylene fumarate graphene oxide nanocomposites (Diez Pascual, ACS Apple Mater 2016) could be included.
Author Response
Please see the attachment.
Kind regards.
Elisabetta Benedetti

Reviewer 2 Report
The authors proposed a niveau and well-documented review.
As far as I’m concern, the figures needs to be well-illustrated but the illustration must be in adéquation with the global idea of the figure. With this in mind, I can’t understand quite well the Figure 1, as well as the acronyms used. Please redo the figure in a less illustrative way.
the same for Figure 2, we can see the limits of all the images used, and the size of the elements inside the circles are too small. Please reconsider this Figure as well.
Author Response

(The authors gave the same response as above.)

Round 2
Reviewer 1 Report
The article has been improved and can be accepted for publication now.
Reviewer 2 Report
The authors revised the manuscript to help the reader to better follow the global ideas developed. Figures are much more informative than in the previous version, and all the complement in the text are appreciated.
This revised version is acceptable for publication.